# Fibroblast-Mediated Immunoregulation of Macrophage Function Is Maintained after Irradiation

**DOI:** 10.3390/cancers11050689

**Published:** 2019-05-17

**Authors:** Rodrigo Berzaghi, Muhammad Asad Ahktar, Ashraful Islam, Brede D. Pedersen, Turid Hellevik, Inigo Martinez-Zubiaurre

**Affiliations:** 1Department of Clinical Medicine, Faculty of Health Sciences, UiT-The Arctic University of Norway, 9037 Tromsø, Norway; rodrigo.berzaghi@uit.no (R.B.); drasadakhtar2@gmail.com (M.A.A.); ashraful.islam@uit.no (A.I.); 2Department of Radiation Oncology, University Hospital of Northern Norway, 9038 Tromsø, Norway; Brede.dille.pedersen@unn.no (B.D.P.); Turid.Hellevik@unn.no (T.H.)

**Keywords:** cancer-associated fibroblasts, CAFs, ionizing radiation, radiotherapy, macrophages, immunoregulation, immunosuppression

## Abstract

The abilities of cancer-associated fibroblasts (CAFs) to regulate immune responses in the context of radiotherapy remain largely unknown. This study was undertaken to determine whether ionizing radiation alters the CAF-mediated immunoregulatory effects on macrophages. CAFs were isolated from freshly-resected non-small cell lung cancer tumors, while monocyte-derived macrophages were prepared from peripheral blood of healthy donors. Experimental settings included both (CAF-macrophage) co-cultures and incubations of M0 and M1-macrophages in the presence of CAF-conditioned medium (CAF-CM). Functional assays to study macrophage polarization/activation included the expression of cell surface markers, production of nitric oxide, secretion of inflammatory cytokines and migratory capacity. We show that CAFs promote changes in M0-macrophages that harmonize with both M1-and M2-phenotypes. Additionally, CAFs inhibit pro-inflammatory features of M1-macrophages by reducing nitric oxide production, pro-inflammatory cytokines, migration, and M1-surface markers expression. Radiation delivered as single-high dose or in fractioned regimens did not modify the immunoregulatory features exerted by CAFs over macrophages in vitro. Protein expression analyses of CAF supernatants showed that irradiated and non-irradiated CAFs produce approximately the same protein levels of immunoregulators. Thus, CAF-derived soluble factors mediate measurable changes on uncommitted macrophages and down-regulate pro-inflammatory features of M1-polarized macrophages. Notably, ionizing radiation does not curtail the CAF-mediated immunosuppressive effects.

## 1. Introduction

Radiotherapy (RT) is a highly efficient and widely used treatment modality for solid tumors and is largely understood as a localized treatment modality, aiming to induce tumor cell killing while keeping effects on the surrounding healthy tissue at a minimum. In recent years, the field of radiation oncology is embracing the notion that RT may trigger not only local but also systemic immune-modulating effects [1,2,3,4]. In fact, preclinical and clinical studies have shown that RT may profoundly alter cellular and acellular components of the tumor microenvironment (TME) [5] and transform it from immunosuppressive to immunoreactive [4,5,6]. Despite the well-described synergism between RT and different forms of immunotherapies, results at the clinical front from radio-immune combinatory trials are still modest [7,8,9,10,11,12]. There are still many overlooked variables in the studies exploring radiation and the immune system. A more thorough biological understanding of the effects elicited by ionizing radiation in TME elements may aid in the design of more efficient treatment strategies.

CAFs represent a heterogeneous population of connective tissue cells that are both numerically and functionally prominent constituents of solid neoplasms [13,14,15]. Numerous correlative studies have demonstrated that a high abundance of tumor-infiltrating CAFs and high stroma/carcinoma ratios is associated with poor prognosis and treatment outcomes [16]. Contrary to quiescent normal tissue fibroblasts, tumor-resident fibroblasts (CAFs) contribute to cancer progression essentially by the secretion of a long array of soluble factors, comprising cytokines, chemokines, growth factors and proteases that on one hand regulate the behavior of adjacent tumor cells [17,18], but also mediate recruitment of inflammatory and immune cells, including mobilization of bone marrow-derived progenitor cells [19]. Amongst the mechanisms that tumors use to evade immune rejection, immunosuppressive effects elicited by non-malignant TME-components play a major role. As a major constituent of the tumor stroma, CAFs participate actively in the regulation of both innate and adaptive anti-tumor immune responses [20]. In fact, through the secretion of a plethora of immunoregulatory signals, stromal fibroblasts are efficient regulators of the local immunity in tumors, with the capacity to directly affect trafficking, state of differentiation and activation of a broad population of immune cells. Among some observed effects, CAFs may influence T-regulatory cells recruitment and functionality by the production of prostaglandin E2 (PGE2) and can impair Th1 differentiation, cytotoxic T lymphocytes (CTL) activation or dendritic cells (DCs) maturation by secretion of transforming growth factor-beta (TGF-β) or the immunoregulatory enzyme indoleamine-2,3-dioxygenase (IDO) [21]. Moreover, some recent reports highlight indirect effects mediated by CAFs on immune cell infiltration and function, consisting of regulation of angiogenesis, lymph-angiogenesis, hypoxia, extracellular matrix remodeling and metabolism [22].

In particular, CAFs may play important roles in the regulation of macrophage recruitment, polarization and functions [23]. Macrophages are broadly classified into two distinct categories: the M1-like or classically activated type and the M2-like or alternatively activated type [24]. M1-macrophages exhibit a pro-inflammatory phenotype characterized by the production of different proinflammatory cytokines and reactive oxygen and nitrogen species and have the capacity to orchestrate Th1 anti-tumor immune responses. In contrast, M2-macrophages orchestrate a tumor-promoting tissue-repair response characterized by the secretion of pro-angiogenic and immunosuppressive factors such as IL-10, IDO, VEGF, TGF-β and Arginase, which inhibit cytotoxic CD8^+^ T-cell mediated immune attack [25]. However, the use of the terms M1 and M2 remains controversial due to lack of tightly defined criteria to score phenotypes [24]. Macrophages display great plasticity and can adopt different activation states within many possibilities. In some settings, CAFs actively promote the migration of monocytes to the TME and their differentiation towards M2-macrophages [26]. Particularly, the secretion of IL-6, CCL2/MCP-1, M-CSF/CSF-1 or SDF-1/CXCL12 by CAFs have been shown to promote migration, cell survival and M2-polarization of monocytes/macrophages [27,28,29].

Ionizing radiation (IR) in a clinical setting affects not only the malignant cells but also all the non-malignant components present in the TME, thus provoking also tumor cell-extrinsic responses to treatment [5,6]. Hence, as a major constituent of the tumor mass, CAFs receive the prescribed radiation dose in full during external beam RT treatment, and the downstream biological effects may depend on radiation dose and fractionation. Evidence from in vitro studies indicates that CAF is a very radioresistant cell type and may survive to even ablative doses of ionizing radiation [30]. However, radiation doses above 12 Gy induce permanent DNA damage responses and the concomitant development of irreversible cellular senescence. Additionally, high radiation doses are able to influence the secretory profile of CAFs, thus affecting potential CAF-mediated paracrine signaling [18]. In this study, we explore if CAF-mediated immunoregulatory effects on macrophages are changed after exposure of CAF cultures to different radiation regimens.

## 2. Results

### 2.1. Isolation of CAFs, Irradiation, and Macrophage Polarization

CAFs were isolated from non-small cell lung cancer tissue by enzymatic digestion and outgrowth, as described by Hellevik et al. [30]. Under the light microscope, CAFs appeared elongated with spindle-shaped morphology, as illustrated in Figure 1A. Fibroblasts expanded in monolayers were checked for lineage-specific markers including alpha-smooth muscle actin (α-SMA) and fibroblast activation protein (FAP-1) [30]. In this study, CAFs were collected from five different donors to complete all the experimentation. Figure 1A illustrates that irradiated cells acquired a flat and enlarged morphology, indicative of prematurely induced cellular senescence and growth arrest. The extent of cell senescence in CAFs was demonstrated by β-galactosidase staining five days after the first radiation dose (Figure 1B). Irradiated cultures showed prominent induction of β-galactosidase staining, suggesting that large numbers of irradiated CAFs entered growth arrest by activating premature cellular senescence mechanisms. The senescence response was more pronounced after a single-high dose (62%) than after fractionated radiation regimens (37%) (Figure 1C). Of note, no cell death was observed after any of the radiation regimens over a 3 weeks period, as assessed by light microscopy (rounded and/or floating cells). Aspects of CAF-viability post-IR have been reported previously by us [31].

Figure 1D shows a representative image of adherent macrophages in co-culture with fibroblasts. Attached fibroblasts exhibited a significantly higher degree of elongation compared with macrophages. M0-macrophages appeared elongated in shape and firmly attached to the surface as compared to the round morphology showed by purified blood (CD14^+^) monocytes (Figure 1E). Addition of LPS and IFN-γ, which stimulate M1 polarization, caused cells to flatten into a round-like shape within 48 h of stimulation.

### 2.2. CAF-Mediated Effects on Macrophage Polarization

To investigate the effects of ionizing radiation on CAF-mediated regulation of macrophage polarization, we exposed unstimulated (M0) or M1-macrophages with CM from irradiated or non-irradiated CAFs or in co-cultures. Thereafter, we analyzed the expression of specific type-1 and type-2 macrophage surface markers by flow cytometry. Considering the potentially different effects triggered by different radiation schemes, we compared the effects of fractioned radiation vs single-high dose radiation on CAFs-mediated macrophage polarization. Results presented in Figure 2, represent average determinations from five different CAFs donors. Figure 2 shows the relative surface expression of M1-macrophage co-stimulatory molecules CD80, CD86, and CD40, as determined by flow cytometry. It is evident that CD80 was significantly enhanced in uncommitted M0-macrophages by all CAF-CMs, but unchanged in stimulated (M1) macrophages (top panel). Neither irradiated (iCAF-CM) nor non-irradiated CAF-CM was able to alter the expression of CD86 in resting (M0) or stimulated (M1)-conditions (middle panel). On the other hand, expression of CD40 was moderately increased by all CAF-CM in M0-macrophages, but was statistically significant (*p* ≤ 0.05) only for the irradiated (CAF-CM) conditions. No effects were observed for CD40 expression in M1-stimulated conditions (lower panel). In co-cultures, CAFs were able to induce a slight increase of CD80 but had no effects on the expression of CD86 or CD40 in M0-macrophages, except for the fractionated irradiated CAF group (fiCAFs), which demonstrated slightly elevated expression of co-stimulatory CD40 (Figure 2). Otherwise, no relevant differences were observed on M1-macrophage marker regulation after exposure to irradiated or non-irradiated CAFs. CAFs grown in co-cultures with M1-macrophages induced a slight decrease of CD80 expression and a significant decrease of CD40 expression, but unaltered effects on CD86 expression. Results were statistically significant (*p* = 0.011) for the fiCAF group regarding CD40 expression levels. Of note, we did not observe statistically significant differences in the expression of any of the receptors in any of the experimental settings when comparing irradiated with non-irradiated CAF conditions both in co-cultures or CM.

In Figure 3, the relative surface expression of the classical M2-macrophage surface markers mannose receptor (CD206) and scavenger receptor (CD163) is illustrated. In non-stimulated conditions (M0), incubation with iCAF-CM or CAF-CM increased the expression of CD206. On the contrary, in stimulated (M1) conditions, the two irradiated CAF-CMs slightly decreased the expression of CD206. Surface expression of CD163 on M0-macrophages increased to some extent after exposure to iCAF-CM and fiCAF-CM, although differences were not significant. Similarly, in M0-macrophages, both irradiated CAF-CMs slightly increased the expression of CD163. No statistical differences were observed between the irradiated and non-irradiated CAF-CM effects on CD163 and CD206 expression on M1-macrophages. In co-cultures consisting of (non-irradiated) CAFs and M0-macrophages, CAFs induced a noticeable enhancement in the expression of M2-markers CD206 and CD163 in the (M0) macrophages. However, this effect was to some extent blocked after co-culturing (M0-) macrophages with fiCAFs or iCAFs. On the other hand, in co-cultures with M1-macrophages and CAFs/iCAFs/fiCAFs, enhanced expression of CD206 was observed from CAFs and iCAFs, whereas enhanced expression of CD163 was only observed from fiCAFs. Nevertheless, differences from controls in these co-cultures were not statistically significant, due to prominent inter-donor heterogeneity.

### 2.3. Effects of CAFs on Macrophage Cytokine Secretion

To further explore the immunoregulatory properties exerted by CAFs, we checked the capacity of CAFs to modulate cytokine secretion by macrophages. To this end, we examined macrophage cytokine production resulting from exposure to CAF-CM or during co-cultures. In Figure 4, results from corresponding analyses of IL-6, TNF-α, IL-10, and IL-12 from uncommitted (M0) and stimulated M1-macrophages is presented. When M0-macrophages were incubated with CAF-CM, all CMs were able to increase, to a certain degree, IL-6 production (Figure 4, top-left panel), reaching statistical significance with the fractioned radiation group (fiCAF-CM) (*p* = 0.014). On the contrary, in the stimulated M1-condition, IL-6 production was unchanged upon exposure to irradiated (CAF-CM) and significantly reduced by CAF-CM. Interestingly, in co-culture conditions that included either M0- or M1-macrophages, IL-6 was significantly elevated in M0-macrophages (CAFs *p* = 0.035 and iCAFS *p* = 0.017) and (or near-significantly) in M1 macrophages from baseline levels by both CAFs and the two iCAFs (Figure 4, top-right panel).

Secreted levels of IL-12 and TNF-α by M0-macrophages was negligible and unchanged in all experimental conditions (Figure 4, middle panels). For M1-macrophages, both iCAF-CM and CAF-CM (*p* = 0.045) resulted in decreased production of IL-12 and reduced TNF-α to some extent. In co-culture conditions, CAFs and iCAFs reduced M1-macrophage secretion of TNF-α and IL-12 (CAFs *p* = 0.093 and iCAFs *p* = 0.068).

Interleukin (IL)-10 is a recognized marker for M2-macrophages [23,32] and expected to be induced under IL-4 and TGF-β stimulation. However, unexpectedly, the production of IL-10 was highest in M1-conditions. Previous studies have also reported similar observations with IL-10 [33]. Conditioned medium from all CAF experimental groups had the capacity to trigger induction of IL-10 secretion by M0-macrophages, and was significantly increased from CAF-CM and iCAF-CM (*p* = 0.035 and *p* = 0.017, respectively) compared to baseline levels (Figure 4, bottom-left panel). IL-10 induction was less prominent with the fiCAF group. On the other hand, exposure of M1-macrophages to iCAF-CM or CAF-CM resulted in reduced secretion of IL-10, with significant reduction upon exposure to fiCAF-CM (*p* = 0.040). Moreover, the CAF-mediated inhibitory effect on IL-10 secretion from M1-macrophages was even more pronounced in co-culture conditions (Figure 4, bottom-right panel).

### 2.4. Effects of CAFs on Macrophage Nitric Oxide Production

Nitric oxide (NO) production is considered a hallmark of macrophage polarization and activation [34]. Therefore, we pursued to measure immunomodulatory effects mediated by CAFs on macrophage nitric oxide production. In Figure 5, monocyte-derived M0- or M1-macrophages were seeded together with CAF-CM or CAFs and incubated for 48 h. Thereafter, NO assay was performed using the membrane permeable fluorescent probe DAF-2DA. Conditioned media from all CAF experimental groups did not change the NO-production by uncommitted (M0) macrophages (Figure 5). On the other hand, in M1-macrophages, CAF-CM, iCAF-CM and fiCAF-CM provoked significant reduction in NO production (*p* = 0.036). However, no differences were observed between CM from irradiated vs non-irradiated CAFs. In co-cultures, both irradiated and non-irradiated CAFs could enhance NO production by M0-macrophages. In contrast, NO production by M1-macrophages was inhibited by all CAF-groups, with significant values only from iCAF (*p* = 0.037).

### 2.5. CAF-Mediated Effects on Macrophage Migratory Capacity

We next sought to explore the capacity of CAFs to modulate the migratory behavior of macrophages. As expected, uncommitted M0-macrophages did not migrate towards CCL19 [35]. However, as illustrated in Figure 6, M1-macrophages exposed to any of the three different CAF-CMs resulted in reduced migratory capacity, with significant values from CAF-CM and or iCAF-CM) (*p* = 0.015 and *p* = 0.003, respectively).

### 2.6. Effects of Radiation on Secretion of CAF-Derived Inflammatory Mediators

CAFs are believed to exert most of their tumor-promoting functions through the secretion of bioactive molecules into the TME, and some of these factors exert demonstrated effects on macrophage polarization and functions [26,28,29]. In this regard, we have explored the impact of radiation on CAFs secretory capacity. A panel of cytokines and chemokines known to regulate macrophage chemotaxis and functions were compared between CAFs/iCAFs/fiCAFs. As is evident from Figure 7, our results showed no significant differences between the CAF-groups regarding cytokine secretion, as determined from our cytokine panel consisting of IL-8, IL-10, VEGF-A, M-CSF, IL-4, TGF-β, CHI3L-YKL-40, CXCL12/SDF-α, IL-6, and CCL2-MCP1. Of note, levels of IL-10 in the conditioned medium from both control and irradiated CAFs were nearly undetectable.

## 3. Discussion

Ionizing radiation leads to significant changes in the tumor stroma, triggering reactions such as acute inflammation, vasculature damage, tumor tissue hypoxia, and fibrosis [5,6,36]. Those changes result in sustained production of cytokines and chemokines that in turn can induce immune cell recruitment and may trigger antitumor immune responses [37]. An array of studies have demonstrated that CAFs and tumor-associated macrophages (TAMs), which together constitute the most abundant non-cancerous cells in the tumor stroma, are synergistically related to cancer progression and prognosis of patients [26,38,39,40]. However, little is known about how radiotherapy is influencing the capacity of CAFs to modulate macrophage polarization in the tumor microenvironment. In the present study, we have investigated (a) how CAFs may affect macrophage polarization and function in vitro; and (b) how ionizing radiation is affecting the immunoregulatory functions exerted by CAFs. Thereafter, we have observed that: (i) CAFs, both in co-culture and by conditioned medium, promote changes on uncommitted macrophages (M0) that harmonize with induction of both M1- and M2-macrophage phenotypes; (ii) CAFs, both in co-culture and by conditioned medium, inhibit some of the pro-inflammatory features of M1-macrophages; and (iii) neither high-dose radiation (1 × 18 Gy) nor medium-high-fractioned radiation (3 × 6 Gy) is affecting considerably the immunoregulatory features exerted by CAFs over macrophages in vitro.

Accumulating research have demonstrated that CAFs play an important role in macrophage infiltration and polarization in different tumors [26,40,41,42]. Recent findings have shown a correlation between CAF abundance and pro-tumoral macrophages infiltration in tumor specimens of patients with colon cancer [39], triple negative breast cancer [41], and oral squamous cell carcinoma (OSCC) [38]. In vitro studies, based on CAFs from various tumor types, support the concept that CAFs preferentially induce the transformation of M0-macrophages into a tumor-promoting (M2) phenotype [26,28,41]. Monocytes co-cultured with CAFs from pancreatic ductal adenocarcinoma have been shown to present with increased surface expression of specific M2-markers, e.g., CD163, CD200R, and CD206 surface receptors and up-regulation of *ARG1*, *IL10*, and *TGFB1* genes [28]. Monocytes grown in co-cultures with OSCC-CAFs have been shown to display increased reactive oxygen species [26]. Of note, Zhang et al. [26] demonstrated that neutralizing antibodies to M-CSF could inhibit ROS-production and M2-polarization of CAF-CM stimulated monocytes. Comito et al. [40] used an in vitro model of prostate cancer and found that CAFs apparently synergize with M2-macrophages to increase cancer cell motility and induce endothelial cells activation and a pro-angiogenic state [40]. Additionally, in a colorectal cancer model, it has been described that CAFs can affect tumor infiltration of inflammatory cells by promoting upregulation of cell adhesion molecules such as VCAM-1 in cancer cells [42]. In this latter model, CAF-derived IL-8 presumably accounted for monocyte recruitment and M2 polarization. In line with previous studies, in our study, we have observed a potent inhibition of M1-signatures on stimulated macrophages exposed to CAFs or CAF-conditioned medium. Moreover, CAFs, irradiated or not, also inhibit the migratory capacity of stimulated M1-macrophages in vitro. On the other hand, unstimulated M0-macrophages, co-cultured with CAFs or CAF-conditioned medium, acquired a heterogenic phenotype with mixed M1-M2 characteristics. Takahashi et al. [28] described similar findings regarding the expression levels of M2-markers and co-stimulatory molecules in monocytes co-cultured with CAFs. Fully polarized M1- and M2-macrophages are only the extremes of a continuum of functional states that cannot easily be binned into an M1 or M2 phenotype [24]. Our results thus suggest that CAF-educated macrophages express a broader transcriptional polarized repertoire that harmonizes with both M1 and M2 phenotypes.

It is now widely acknowledged that CAFs represents a heterogeneous population of mesenchymal cells that can differ among tumor type, tumor stage and tumor location in the body [43]. The pleiotropic nature of CAFs can associate with both pro-inflammatory and anti-inflammatory gene signatures and the secretion of soluble signal mediators, e.g., TNF-α, IFN-γ, TGF-β, PGE2, IDO, CXCL-12/SDF1, CCL2/MCP-1, Tenascin-C, IL-6, IL-4, CXCL8 [17], and nitric oxide [44], that potentially affect macrophage polarization in different ways at different stages of cancer progression [44]. Collectively, our results propose that CAFs promote changes in the phenotype of uncommitted macrophages, inducing both M1- and M2-type characteristics. We have also confirmed that CAFs have a role in redirecting M1-macrophages to an M2-phenotype, as indicated by strong inhibition of pro-inflammatory cytokines secretion, nitric oxide production, and migration capacity.

Furthermore, we assessed the effect of fractioned versus single high-dose radiation on CAF-mediated macrophage polarization. Previous studies, by us and others, have presented CAFs as a highly radioresistant cell type that can sustain ablative radiation doses [30,45]. However, CAFs exposed to a single high-dose of 12 Gy (and above) develop significant morphological changes associated with premature cellular senescence, and an altered phenotype accompanied by functional traits as reduced proliferation, migration, and invasion rates [30]. Importantly, irradiated CAFs have also been shown to display increased pro-tumorigenic effects in both in vitro and in vivo models [46,47,48]. Although radiotherapy induces profound phenotypic and secretory changes on CAFs, a previous study from our group suggests that neither high-dose nor low-dose RT affect CAF-mediated immunosuppressive functions over lymphocytes in vitro [49]. Likewise, experimental in vivo studies have shown comparable amounts of tumor-infiltrated macrophages in mice subcutaneously inoculated with mixed tumor cells and irradiated or non-irradiated CAFs [31].

In the study presented here, the two radiation schemes induced morphological changes on CAFs, but with an increased rate of senescence in CAF cultures submitted to a higher radiation dose. However, none of the two radiation regimens were able to modify markedly the regulatory functions of CAFs over macrophages. In agreement with this observation, radiation did not affect the release of relevant soluble immunomodulators by CAFs.

Macrophage phenotype and recruitment can also be directly affected by radiation. Recent studies have shown that fractionated medium-doses, between 2 and 5 Gy, is able to reprogram macrophages from M2- to the pro-immunogenic M1-phenotype in vitro and in vivo [50,51]. In addition, high-dose radiation of 10 Gy or higher induces the release of damage-associated molecule patterns (DAMPs), such as high mobility group box 1 (HMGB1), by apoptotic and necrotic cells and induces the recruitment of CD11b+ myeloid cells in the TME and reprogramming macrophages towards the tumor-promoting M2-phenotype [52,53,54]. According to the presented data, regardless of the radiation scheme, RT-induced senescent CAFs may still participate in the recruitment of myeloid and other immune suppressor cells. Moreover, such senescent CAFs may promote the transformation of macrophages towards a tumor-favoring phenotype. In our study, we have not tested radiation doses under 6 Gy, which in the context of tumor macrophages could be relevant as small doses in the range of 2 Gy favor the appearance of anti-tumoral M1 macrophages.

## 4. Materials and Methods

### 4.1. Human Material, CAF Isolation, and Cultures

Human lung CAFs were isolated from freshly resected non-small cell lung carcinoma (NSCLC) tumor tissue from patients undergoing surgery at the University Hospital of Northern Norway (UNN), Tromsø, as previously described [27]. Lung tumor specimens from five different patients and blood (i.e., buffy-coats) from 10 unrelated healthy donors were used in the study, under patient written informed consent. All methods involving human material were performed in accordance with relevant ethical guidelines and regulations. The Regional Ethical Committee of Northern Norway has approved the use of human material that has been included in this study (REK Nord 2014/401; 2016/714; 2016/2307).

NSCLC-derived CAFs were isolated by enzymatic digestion of tissues and the outgrowth method and characterized by the presence of lineage-specific markers, as described previously [27]. Isolated CAFs were cultivated in Dulbecco’s modified Eagle’s medium (DMEM) (Cat. # D6046 Sigma-Aldrich, St Louis, MO, USA) supplemented with 10% fetal bovine serum (FBS) (Cat. # S0115 Biochrom, Berlin, Germany) and used for experimentation after the third and fourth passage (3–4 week-old cultures).

### 4.2. Irradiation of Cells

Adherent CAF cultures—cultivated in DMEM with 10% FBS and grown in T-175 flasks or in 24 well culture plates—were irradiated when 70–90% confluent with high-energy photons, using a clinical Varian linear accelerator, as previously described [27]. Ionizing radiation was delivered to cultured cells either as a single-high dose (1 × 18 Gy) or in fractionated schemes (3 × 6 Gy) at 24 h intervals. Standard parameters for dose delivery was depth 30 mm, beam quality 15 MV, dose rate of 6 Gy/min and field sizes of 20 × 20 cm.

### 4.3. Beta-Galactosidase Assay

Five days post-irradiation, CAF cultures (20 × 10^5^ cells/well in 6-well plates) were washed and fixed (5–7 min) at room temperature with freshly prepared paraformaldehyde (PFA, 2%). β-galactosidase (5-bromo-4chloro-3-indolyl-B-D-galactopyranoside) staining was performed following instructions from the manufacturer (Cat.no CS0030, Sigma-Aldrich, St. Louise, MO, USA). Quantity of senescent cells in each experimental group was determined by counting blue cells on minimum three randomly selected fields under a Nikon Eclipse TS100 model light microscope (Tokyo, Japan). Randomly selected fields were photographed at 100× magnification, using an Idea SPOT digital camera.

### 4.4. CAF-Conditioned Medium

At the third passage, CAFs were seeded at a density of 4 × 10^5^ cells per T-75 tissue culture flask and incubated (24 h, 37 °C) in DMEM (with 10% FBS). After the initial cell attachment and spreading, cultures were gently washed with PBS (37 °C), new incubation medium (6 mL) was added, followed by irradiation of cultures, as previously described [27]. Culture medium from CAFs exposed to ionizing radiation (3 × 6 Gy) was conditioned for 48 h after the last radiation dose. For the group exposed to (1 × 18 Gy), CM has been collected between day 3 and day 5 after irradiation. Culture supernatants were then spun down by centrifugation (2000× *g*, 4 °C, 10 min) and then filtrated (Ø = 0.45 µm) for elimination of contaminant cell debris. The resulting conditioned medium was either used immediately for experimental analysis or frozen at −80 °C for later use.

### 4.5. Isolation of Peripheral Blood Mononuclear Cells and Generation of Macrophages

Peripheral blood mononuclear cells (PBMCs) were isolated from human blood (i.e., buffy-coats) using Lymphoprep-TM (StemCell Technologies, Vancouver, BC, Canada) gradient centrifugation. From the selected (PBMC)-pool, CD14^+^ monocytes were isolated using magnetic CD14^+^ Microbeads (Cat. no. 130-050-201; Miltenyi Biotec, Bergisch Gladbach, Germany). Purity and recovery of CD14^+^ monocytes were determined by CD14-FITC antibody labeling (Cat. no. 130-113-708; Miltenyi Biotec), whereas cell viability was determined by propidium iodide (PI) staining and subsequent flow cytometry analysis on a BD FACSAria III (BD Biosciences, San Jose, CA, USA). For the generation of macrophages, CD14+ monocytes were cultured in macrophage growth medium (RPMI 1640 with 10% FBS, 1% streptomycin/penicillin and 100 ng/mL of macrophage colony-stimulating factor (M-CSF); (Cat. no. 300-25; PrepoTech, Rocky Hill, NJ, USA) and kept in a humidified atmosphere (5% CO_2_, 37 °C), for 6 days. The incubation medium was replaced every three days with fresh (pre-warmed) medium.

### 4.6. Macrophage Polarization Protocols

M1-differentiation of uncommitted macrophages (M0) were effectuated following previously published protocols [28]. Briefly, M1-polarization was attained by incubating M0-macrophages in medium containing a mixture of LPS (100 ng/mL) (Cat. no. L6529; Sigma Aldrich, St. Louis, MO, USA) and IFN-γ (20 ng/mL) for 48 h at 37 °C (Cat. no. 300-02; PrepoTech-USA). After 48 h of incubation with stimulants, M1- and M2-macrophages were phenotypically characterized through surface markers expression, nitric oxide production and the release of various inflammatory cytokines.

### 4.7. Cell Co-Culturing and Macrophage Stimulation by CAF-Conditioned Medium

In co-culture experiments, irradiated and non-irradiated CAF cultures were established in 24-well plates (2 × 10^5^ cells per well). M0-macrophages were thereafter added at a density of 4 × 10^5^ live cells per well (Macrophages/CAFs ratio; 2:1). These cultures, with mixed cell types, were further incubated for 48 h at 37 °C in macrophage growth medium (RPMI 1640 with 10% FBS, 1% streptomycin/penicillin). Similar procedures were carried out for experiments with CAF-CMs, but instead of cells (CAFs), CAF-CM was diluted (1:1) with fresh pre-warmed macrophage growth medium and added to the macrophage cultures. In M1-macrophage conditions, immediately after initiation of co-culture (macrophages and CAFs cells) or macrophages cultured with CAF-CM, cells were exposed to LPS (100 ng/mL) and IFN-γ (20 ng/mL). Following (cellular co-culturing or CAF-CM) incubations, macrophages were harvested and used for further analysis.

### 4.8. Macrophage Cell Surface Markers by Flow Cytometry

The expression of surface markers on macrophages was analyzed by flow cytometry on BD FACSAria III using the FlowJo software, Ver.7.2.4 (Tree Star, Ashland, OR, USA). Briefly, macrophage preparations (2.5 × 10^5^ cells/condition) were labeled with various fluorescent antibodies specific for each phenotype (Miltenyi Biotec, Bergisch Gladbach, Germany). Hence, M1-phenotype markers consisted of CD40, CD80, CD86 (Cat. no. 130-099-385, 130-110-371 and 130-113-571, respectively) whereas M2-phenotype was identified by the markers CD206 (mannose receptor) and CD163 (Cat. No. 130-095-131 and130-099-969, respectively). Isotype controls consisted of mouse IgG1 (Cat. no. 130-098-845), REA control (Cat. no. 130-104-612) and mouse IgG2a (Cat. no. 555574; BD Biosciences-USA). In co-culture experiments, the macrophage population was gated based on the surface expression of CD45 proteins (Cat. no. 130-110-635).

### 4.9. Macrophage Nitric Oxide Production

Intracellular production of nitric oxide in macrophages was analyzed by using a cell-permeable diacetate derivative of 4,5-Diaminofluorescein (DAF-2 DA; Cat. no. 251506; Merck Sharp and Dohme, Kenilworth, NJ, USA). DAF-2-DA dissolved in RPMI (without phenol red; Cat. no. 11835-030; ThermoFisher Scientific, Waltham, MA, USA) penetrates plasma membranes rapidly (at 37 °C) and is hydrolyzed in the cytosol by intracellular esterase activity to DAF-2, which in turn reacts with NO (produced by NOS) to form a soluble fluorescent triazolofluorescein. Briefly, 1 × 10^6^ macrophages were incubated with DAF-2DA (10 μM) in RPMI (without phenol red) for 30 min, at 37 °C. After removing the DAF-2DA-containing medium, cells were washed (3×) with cold PBS and analyzed on a FACSAria III flow cytometer. The fluorescent product was measured using excitation wavelength 450−495 nm and emission wavelength 505–550 nm. A total of 30,000 events were saved and analyzed using the FlowJo software, Ver.7.2.4 (Tree Star, Ashland, OR, USA). Changes in fluorescence intensity (MFI) were calculated by subtracting intensity numbers by unstained cells from stained cells.

### 4.10. Macrophage Migration

CCR7-dependent migration of M1-macrophages towards CCL19 was measured using a Boyden chamber assay. Briefly, 200 μL of M1-macrophages in suspension (5 × 10^5^ cells/mL), previously incubated with CAF-CM during 48 h (as described above), was added to the top chamber of Transwell culture inserts (Ø = 6.5 mm, pore size 8 μm, Cat. # CLS3464, Sigma-Aldrich, St. Louis, MO, USA). Bottom chambers were filled with fresh standard fibroblast growth medium (600 μL) or with CM from irradiated and non-irradiated CAF cultures, in the presence or absence of CCL19 (50 ng/mL) (Cat. # 130-105-744, Miltenyi Biotec, Bergisch Gladbach, Germany) and placed in the upper compartment of a Transwell Plate. After incubation in a humidified atmosphere (5% CO_2_, 37 °C, 3 h), macrophages that had migrated into the lower compartment were harvested and counted under light microscopy.

### 4.11. Macrophage Cytokine Secretion

Quantitative determinations of protein levels of TNF-α, IL-6, IL-10 and IL-12 (Cat. no DY240-05, DY206-05, DY217B-05, and DY1240-05, respectively) in macrophages culture medium, were determined using ELISA kits (R&D Systems, Minneapolis, MN, USA) according to the manufacturer’s instructions.

### 4.12. Multiplex Protein Arrays

A panel of nine-9 specific proteins, including cytokines and chemokines, was measured in the irradiated (1 × 18 Gy and 3 × 6 Gy) or non-irradiated CAF-CM from five different donors, by immune-based protein arrays. A customized human cytokine Multiplex kit (Cat. no. LXSHM-09; R&D Systems, Minneapolis, MN, USA) was used to define the concentration of four-4 different cytokines including M-CSF, IL-4, IL-6, and IL-10; three-3 chemokines including CCL2, CXCL12, CXCL8; one-1 growth factor i.e., VEGFA; and one-1 glycoside hydrolase i.e., CH13L1. All samples were analyzed in duplicates and in 1:2 or 1:4 dilutions. Quantitative protein measurements (two replicates per factor) were performed by using the Luminex Bio-Plex 200 system (Bio-Rad, Hercules, CA, USA). Measured protein concentrations were normalized with cell numbers at specific culture conditions and expressed as pg/mL/10^6^ cells.

### 4.13. Statistical Analysis

All statistical analyses were performed using IBM SPSS statistics version 25 (IBM, Chicago, IL, USA). Comparison of data between the three experimental groups was analyzed using the non-parametric Kruskal-Wallis test, and significance values were adjusted by Bonferroni correction for multiple comparisons. The level of significance was set at *p* < 0.05. Results were presented in graphs, where each donor was plotted as an individual dot in the dataset. In Multiplex protein arrays and ELISAs, only readings above the detection limit of the assay are represented in figures. The β-galactosidase assay was analyzed using Student’s *t*-test and *p*-values were showed in the graphic.

## 5. Conclusions

Taken together, our data display that CAF-derived soluble factors mediate measurable changes on uncommitted macrophages and also down-regulate pro-inflammatory features of M1-polarized macrophages. Notably, our results indicate that ionizing radiation does not curtail the CAF-mediated immunosuppressive effects.

## Figures and Tables

**Figure 1 cancers-11-00689-f001:**
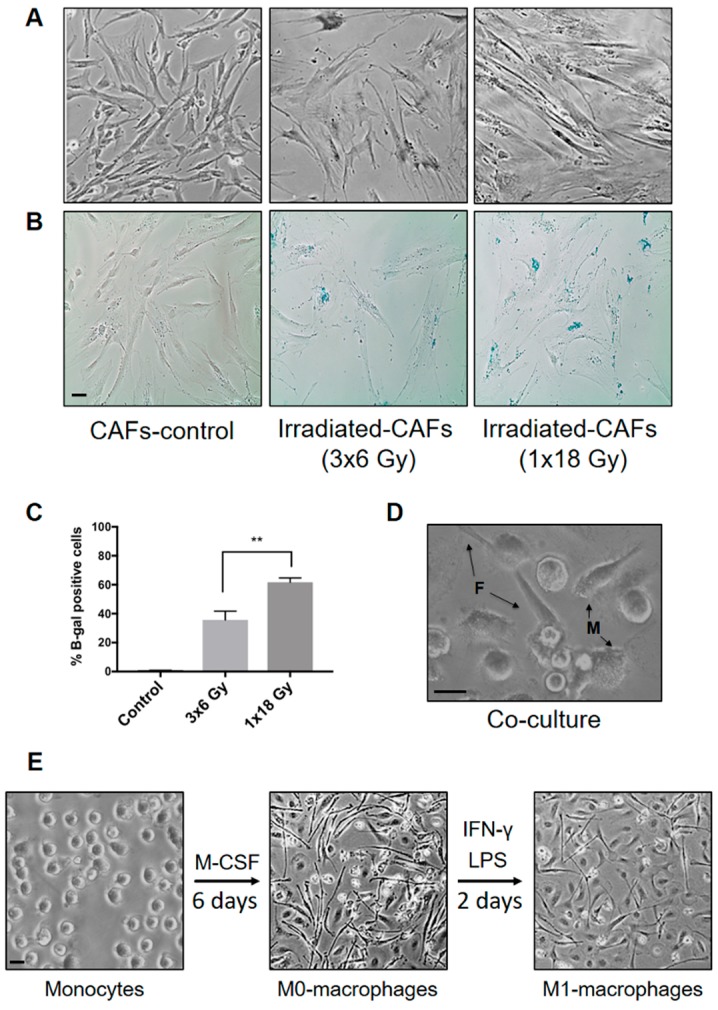
(**A**) Cancer-associated fibroblasts (CAF) cultures and senescence induction by ionizing radiation. Culture-expanded human lung tumor CAFs were divided into three groups and irradiated with a clinical linear accelerator. One group was irradiated once with high-dose radiation (1 × 18 Gy) and another with three daily doses of 6 Gy. (**B**) Radiation-induced senescence was demonstrated by a β-galactosidase assay, where β-gal positive cells develop blue color. (**C**) Percentage of β-galactosidase positive cells in each condition was calculated from two different donors. Student’s *t*-test and *p*-values were determined between the two iCAF-groups. (**D**) Photomicrograph showing CAFs and macrophages in co-culture. (**E**) Macrophage isolation and polarization: Monocytes (CD14^+^) were isolated from PBMCs and incubated with M-CSF to induce macrophage differentiation (M0-phenotype). Monocyte-derived M0-macrophages were further polarized into M1-macrophages by LPS and IFN-γ. Scale Bars = 15 μm.

**Figure 2 cancers-11-00689-f002:**
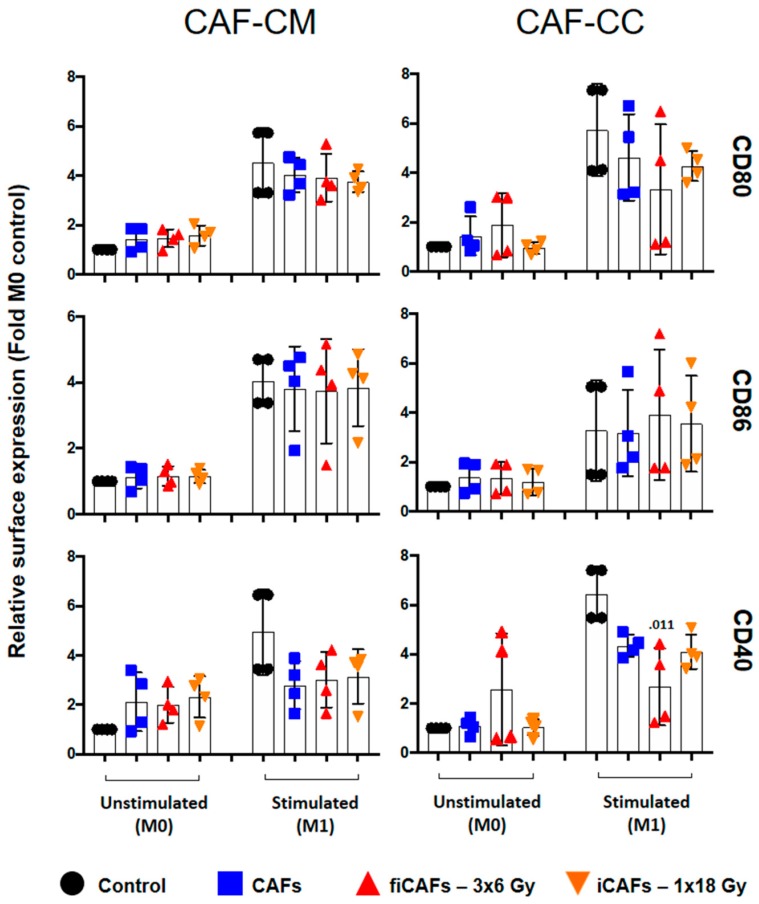
Effects of CAFs on M1-macrophage cell surface markers. Monocyte-derived macrophages in non-stimulated or stimulated conditions were incubated with conditioned medium from irradiated (iCAF-CM) or non-irradiated CAFs (CAF-CM) or in (macrophage-CAF) co-cultures (CC). Resulting surface expression of M1-macrophage cell surface markers CD80, CD86 and CD40 were evaluated by flow cytometry. Filled black circles indicate surface levels in control macrophage cultures (M0 and M1). Results are expressed as fold M0-controls. Data represent mean (±SD) values from four-4 different CAF donors measured independently. Non-parametric Kruskal-Wallis test and *p*-values were determined between control and non-irradiated CAFs, control and the two iCAF-groups individually. iCAF (irradiated CAFs); fiCAFs: fractionated-irradiated CAFs.

**Figure 3 cancers-11-00689-f003:**
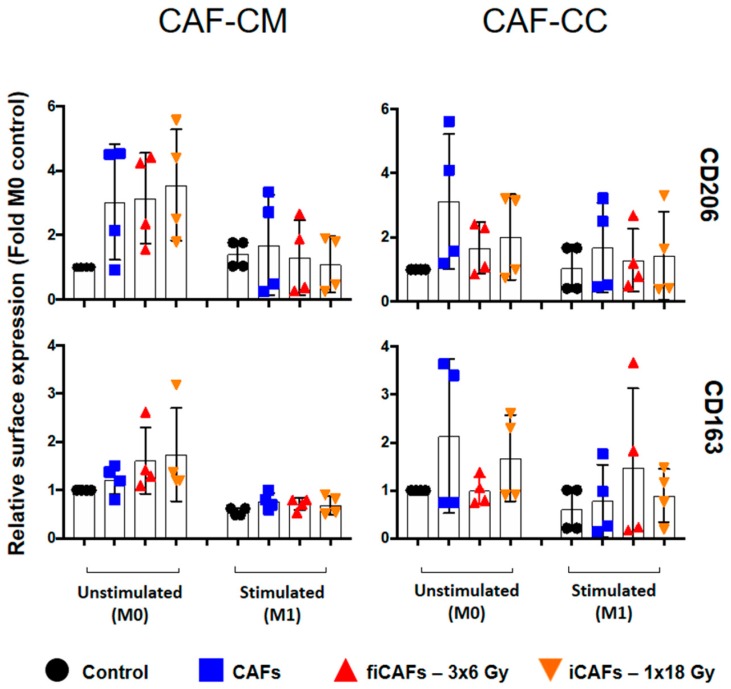
CAF-mediated effects on M2-macrophage cell surface markers. Monocyte-derived macrophages in non-stimulated or stimulated conditions were incubated with irradiated and non-irradiated CAF-conditioned medium (CM) or in co-cultures (CC). Resulting expression of M2-macrophage cell surface markers CD206 and CD163 was evaluated by flow cytometry. Filled black circles indicate expression levels in control macrophage cultures (M0 and M1). Results are expressed as fold M0-controls. Data represent the mean (±SD) values from four-4 different CAF donors in conditioned medium experiments and three donors in co-culture experiments were measured independently. Non-parametric Kruskal-Wallis test and *p*-values were determined between control and non-irradiated CAFs, control and the two iCAF-groups individually. iCAF (irradiated CAFs); fiCAFs: fractionated-irradiated CAFs.

**Figure 4 cancers-11-00689-f004:**
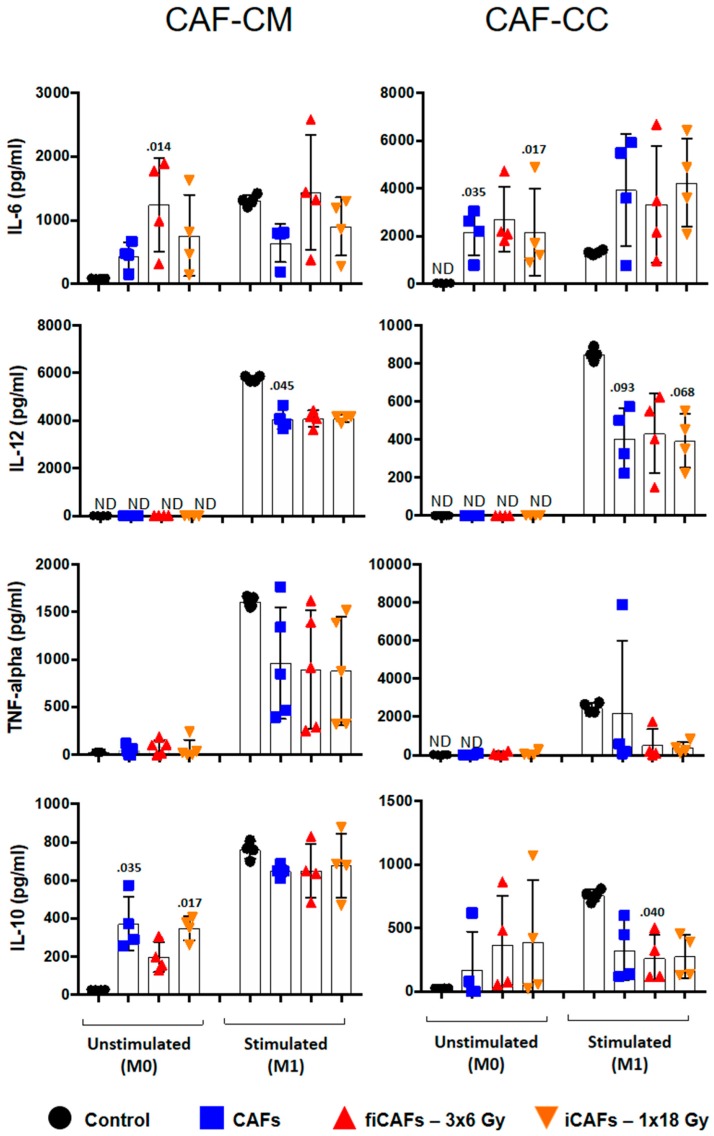
CAF-induced cytokine secretion by macrophages. Monocyte-derived macrophages in non-stimulated or stimulated conditions were incubated with irradiated or non-irradiated CAFs conditioned medium (CM) or in (CAF-macrophage) co-cultures (CC). Resulting levels of cytokines secreted by macrophages were quantified by ELISA assay. Data represent the mean (±SD) values from four different CAF donors measured in duplicates. CAF-derived IL-6 was subtracted from full measurements. Non-parametric Kruskal-Wallis test and *p*-values were determined between control and non-irradiated CAF-CM, control vs iCAF-CM or fiCAF-CM.

**Figure 5 cancers-11-00689-f005:**
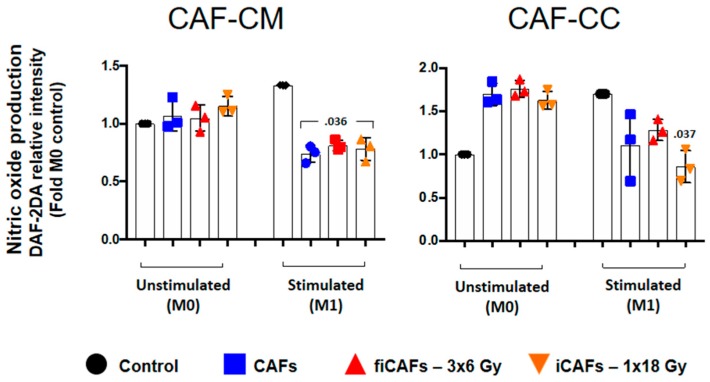
CAF-induced nitric oxide production by macrophages. Monocyte-derived macrophages in non-stimulated or stimulated conditions were incubated with irradiated or non-irradiated CAF-conditioned medium (CM) or in co-culture (CC), followed by analysis of expression levels of nitric oxide by macrophages, measured by DAF-2DA fluorescence assay. Black filled circles indicate expression levels in control macrophage cultures (M0 and M1). Results are expressed as fold M0-controls. Data represent the mean (±SD) values from 3 different CAF donors measured independently. Non-parametric Kruskal-Wallis test and *p*-values were determined between control and non-irradiated CAFs, control and (fractionated) irradiated CAFs individually.

**Figure 6 cancers-11-00689-f006:**
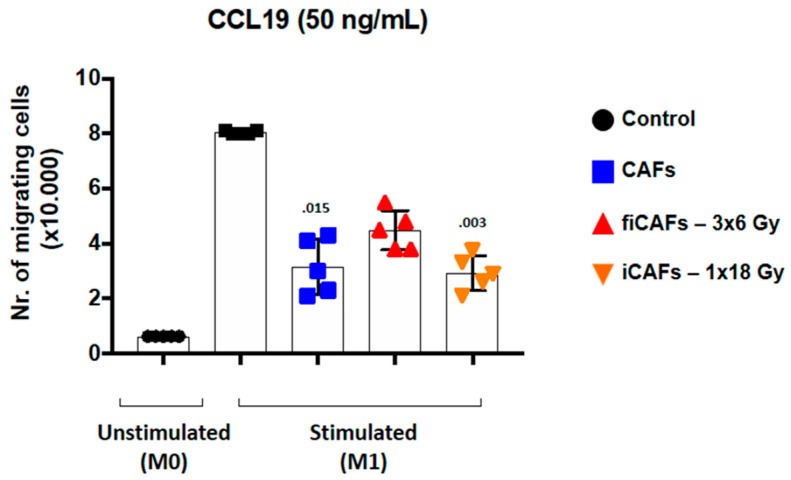
CAF-mediated effects on macrophage migration. For the Transwell migration assays, stimulated macrophages were initially incubated (48 h) with CM from irradiated or non-irradiated CAFs before seeding onto the insert membrane. The migration index was determined by counting transmigrated cells towards CCL19, which was added to the lower chamber. Non-parametric Kruskal-Wallis test and *p*-values were determined between control and non-irradiated CAF-CM, control and the two groups of iCAF-CM. Error bars indicate ± SD of five different CAF donors from triplicate determinations.

**Figure 7 cancers-11-00689-f007:**
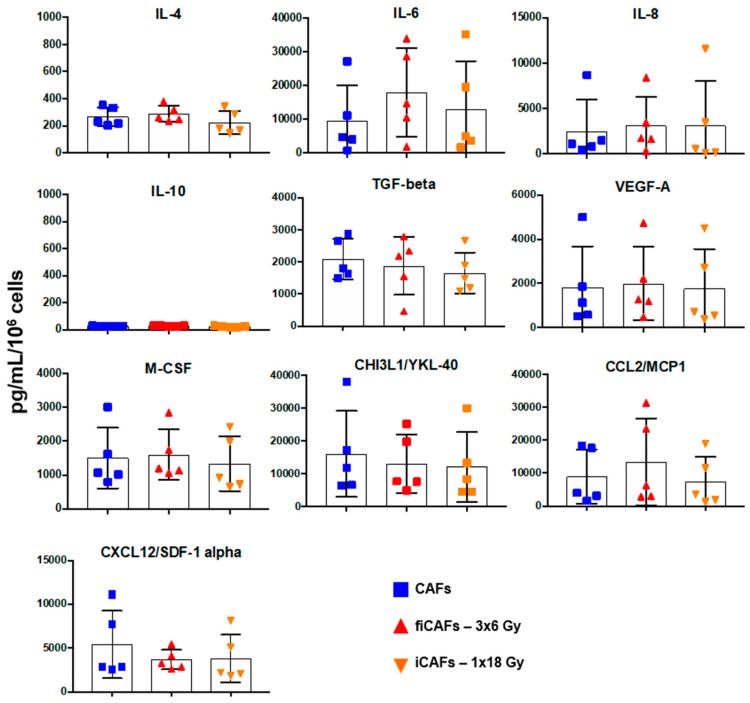
Effects of radiation on CAF-mediated secretion of inflammatory mediators. The release of various macrophage regulatory factors from CAFs was measured in the various CAF-CMs (5 days post-IR) by panels of multiplex protein arrays and ELISA. Results are expressed as fold CAF-control. Mean values from five different CAF donor samples are shown.

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
