# Peer review of "Fibroblast-Mediated Immunoregulation of Macrophage Function Is Maintained after Irradiation"

_cancers, 2019, doi:10.3390/cancers11050689_

Round 1

Reviewer 1 Report

The authors examined if ionizing radiation is altering the CAF-mediated immunoregulatory effects on macrophages. And they found even after radiation CAF continuously control M1 condition without big difference.

It is unclear how many healthy donors or patients contributed to supply monocytes or CAF for the research and whether those monocytes and CAF are mixed together? The author should describe clearly. 

The authors should show survival rate of CAF cells after radiation.

In fig. 4 there some difference in TNFalfa concentration between in radiation plus and minus condition. The authors should describe the possibility more extensively.  I hope they can add more detail data or at least they discuss about the point.   

I think the authors can improve the manuscript in same content.  

Author Response

Reviewer 1.

The authors examined if ionizing radiation is altering the CAF-mediated immunoregulatory effects on macrophages. And they found even after radiation CAF continuously control M1 condition without a big difference.

1-      It is unclear how many healthy donors or patients contributed to supply monocytes or CAF for the research and whether those monocytes and CAF are mixed together? The author should describe clearly. 

Our answer: In this study, we used lung tumor specimens from five different donors as mentioned in the materials and methods section (highlighted in the text - line 361). For monocyte isolation, we have used blood from 10 healthy unrelated donors (highlighted in the text - line 362). Also, in the same section, we added new text “Cell co-culture and CAFs conditioned medium stimulation” (lines 418 to 429), where we clarify the methodology regarding co-culture of macrophages with CAFs and CAFs-conditioned medium.

2-      The authors should show the survival rate of CAF cells after radiation.

Our answer: We have previously shown data regarding this point raised by the reviewer (Grinde et al. Ionizing radiation abrogates the pro-tumorigenic capacity of cancer-associated fibroblasts co-implanted in xenografts. Sci Rep. 2017). In the results section we make a remark on the survival rate of CAF cells after radiation exposure (highlighted in the text - lines 105 to 107).

3-      In fig. 4, there some difference in TNF-alfa concentration between in radiation plus and minus condition. The authors should describe the possibility more extensively. I hope they can add more detail data or at least they discuss the point.

Our answer: TNF-alfa results in figure 4 show a blocking effect when macrophages are treated with CAF-CM or in co-cultures. Differences are not statistically significant when comparing those groups to non-treated macrophages because of big inter-donor variability. However, in contrast to the statement made by the reviewer, we found no differences between irradiated and non-irradiated CAF groups.

Reviewer 2 Report

Authors focus on interaction between carcinoma associated fibroblasts (CAFs) and macrophages, where they distiguish the effects on M0, M1 and M2 macrophages. Authors performed protein expression analyses of CAF supernatants and showed that irradiated and non-irradiated CAFs produce approximately the same protein levels of immunoregulators, as well as, CAF-derived soluble factors mediate measurable changes on uncommitted macrophages and down-regulate pro-inflammatory features of M1-polarized macrophages both in irradiated and non-irradiated conditions.  

The Introduction of the manuscript contains all relevant information and perfectly introduces the issues on radiation effects on CAFs, immune and other stroma cells, the polarisation process and plasticity of the macrophages and current relevant topics on tumor immunology.

The methods are described in appropriate details, the results are presented in sufficient quality and the Discussion also perfectly covers the novel results and also the relevant literature.

One minor comment to the submitted manuscript:

Please also mention the effect of radiated epithelial cancer cells, especially the ones going to necrosis on the three classes of macrophages and compare this with the new observed effects of scenescent irradiated CAFs.

Author Response

Reviewer 2.

Authors focus on the interaction between carcinoma-associated fibroblasts (CAFs) and macrophages, where they distinguish the effects on M0, M1 and M2 macrophages. Authors performed protein expression analyses of CAF supernatants and showed that irradiated and non-irradiated CAFs produce approximately the same protein levels of immunoregulators, as well as, CAF-derived soluble factors mediate measurable changes on uncommitted macrophages and down-regulate pro-inflammatory features of M1-polarized macrophages both in irradiated and non-irradiated conditions.  

The Introduction of the manuscript contains all relevant information and perfectly introduces the issues on radiation effects on CAFs, immune and other stromal cells, the polarisation process and plasticity of the macrophages and currently relevant topics on tumor immunology.

The methods are described in appropriate detail, the results are presented in sufficient quality and the Discussion also perfectly covers the novel results and also the relevant literature.

One minor comment to the submitted manuscript:

1-      Please also mention the effect of radiated epithelial cancer cells, especially the ones going to necrosis on the three classes of macrophages and compare this with the new observed effects of senescent irradiated CAFs.

Our answer: This is a relevant point raised by the reviewer. New text on the effects of radiation on macrophage phenotype and recruitment has been added in the discussion part of the manuscript (highlighted in the text - lines 345 to 356).